# Biomolecules of Fermented Tropical Fruits and Fermenting Microbes as Regulators of Human Hair Loss, Hair Quality, and Scalp Microbiota

**DOI:** 10.3390/biom13040699

**Published:** 2023-04-20

**Authors:** Wolfgang Mayer, Michaela Weibel, Chiara De Luca, Galina Ibragimova, Ilya Trakhtman, Zaira Kharaeva, Danny L. Chandler, Liudmila Korkina

**Affiliations:** 1Medena AG, 16 Industriestrasse, CH-8910 Affoltern-am-Albis, Switzerland; wolfgang.mayer@medena.ch (W.M.); michaela.weibel@medena.ch (M.W.); chiara.deluca@medena.ch (C.D.L.); 2Centre for Innovative Biotechnological Investigations Nanolab (CIBI-NANOLAB), 197 Vernadskiy Pr., 119571 Moscow, Russia; nint2000@yandex.ru; 3Swiss Dekotra GmbH, 549 Badenerstrasse, CH-8048 Zurich, Switzerland; trakhtman@dekotra.com; 4Department of Microbiology, Virology, and Immunology, Kabardino-Balkar Berbekov’s State University, 176 Chernishevskiy St., 360000 Nal’chik, Russia; irafe@yandex.ru; 5Dolsan AG, 6B Bergrain, CH-8910 Affoltern am Albis, Switzerland; chandler@dolsan.ch

**Keywords:** alopecia, ATP, caffeine, clinical trial, fermented papaya/mangosteen, hair loss, pre-/probiotics, MDA, skin microbiota, SH-groups

## Abstract

Plant-derived secondary metabolites (polyphenols/terpenes/alkaloids) and microbial exometabolites/membrane components of fermented tropical fruits are known as highly bioavailable biomolecules causing skin and hair improvement effects (wound healing, anti-inflammatory, antioxidant, antidiabetic, antiacne, skin/hair microbiota balancing, hair growth-promoting, and hair loss-inhibiting). Caffein is considered as a hair growth promoter. A randomized placebo- and caffein-controlled clinical trial on the efficacy of fermented papaya (FP) plus fermented mangosteen (FM) towards human hair quality and loss was conducted. Shampoo and lotion hair care products containing FP, FM, and caffein as active agents were developed and applied to 154 subjects of both sexes with clinically confirmed androgenic or diffuse alopecia for 3 months. Their clinical efficacy was assessed subjectively by questionnaires filled in by dermatologists/trichologists, and by the objective trichomicroscopical calculations. Hair and scalp skin quality was determined by microbiota pattern and ATP, SH-groups, protein, and malonyl dialdehyde quantification. Comparative clinical data showed that the experimental hair care cosmetics significantly inhibited hair loss, increased hair density/thickness, and improved hair follicle structure versus placebo and caffein controls. The cosmetics with FP and FM substantially normalized the microbiota pattern and increased ATP content in hair follicle, while inhibiting lipid peroxidation in the scalp skin, and SH-group formation in the hair shaft.

## 1. Introduction

Hair loss (alopecia) is a widespread health problem affecting many people and thus causing an immense medical and psychological burden that deteriorates the patient’s quality of life. Very often, alopecia reflects existing disorders in humans—hormonal status, reactions to psychological and environmental stresses, and to pharmacological treatments. Sometimes, the etiology of abnormal hair loss remains unknown. The status of scalp skin and quality of hair has been regarded as an important risk factor for the pathogenesis of alopecia [1].

Regarding androgenetic alopecia, which affects both men and women, the treatments are numerous and diverse, including hair transplantation, paramedical physical devices, cosmetics, and pharmacology. Current pharmacological therapies of androgenetic alopecia based on FDA-approved drugs are few, with topical minoxidil (2,4-di-amino-6-piperidinopyrimidine-3-oxide) [2,3] and oral finasteride/dutasteride as active substances [4,5,6,7]. Combinations of oral and topical therapies have been evaluated as well [8,9]. These drugs prevent and cure androgenetic alopecia by inhibiting the production and/or metabolism of sexual hormones (testosterone and estrogens) [10]. Since hair loss requires long-term treatment, these drugs exhibit undesirable side effects [11]. As target cells, dermal papilla cells have been identified in the in vitro cultural experiments, showing that minoxidil promoted cell proliferation while inhibiting their death by apoptosis [2]. Animal experiments confirmed also the activation of the hair follicle epithelial stem cells as the leading mechanism of the hair growth promotion by minoxidil [3]. The adenosine-3-phosphate (ATP)-dependent potassium channels seem to be the major molecular target for minoxidil [12].

Immunosuppressive drugs represent leading therapeutic options for the treatment of alopecia areata [13]. Several nonsurgical physical treatments, such as low-level laser therapy, have also been introduced [6,11]. Autologous cell-based approaches, such as platelet-rich plasma and adipocyte-derived stem cells have been developed recently [14]. However, their clinical efficacy and safety have not yet been proven. Therapies for diffuse hair loss (*telogen effluvium*) of unknown etiology, or connected with chemo-/radiotherapies of tumours, or associated with environmental hazards or psychological stress are scarce and contradictory. Moreover, drugs-inducers of hair growth proliferation (hormones, growth factors, or low molecular weight substances—agonists of Wnt/β-catenin pathway stimulating hair growth [15]), such as valproic acid and lithium chloride [14,16], are all strongly considered to be incompatible with chemotherapy due to the risk of cancer cell growth stimulation [17].

Nowadays, arachidonic acid and its major metabolites prostaglandins, mainly prostaglandin F2α, have been attentively studied as prospective remedies against alopecia (reviewed in [14]), since they promote hair shaft elongation and proliferation of keratinocytes in the hair follicle cultures [18], and have positive clinical outcomes [19]. Unfortunately, there are serious concerns about both safety and patients’ compliance with these treatments.

Owing to these unmet clinical needs, the extensive search for clinically efficient and safe (adverse effects-free) pharmacological/physical means for the therapy and prevention of alopecia has been constantly growing. In addition, the development of hair care cosmetics to normalize scalp skin status and ameliorate hair shaft and root quality has become a great priority.

Recent advances in understanding hair biology and pathology have encouraged the research and development of novel and safer natural hair growth agents. As a common current trend, natural plant-derived or plant substances-based remedies have flooded the immense and fast-growing market of hair care and anti-hair loss paramedical and cosmetic products. The number of publications describing the discovery, development, and research on the mechanisms of the anti-hair loss and pro-hair growth effects of natural herbal/marine phytoextracts, or of their isolated substances has been growing exponentially. The dedicated comprehensive reviews of analytical, preclinical, and clinical data are many [13,20,21], even if the balance of phytochemical and preclinical in vitro/in vivo publications is in great advantage as compared to clinical works. Many herbal-derived actives and their combinations have been used in folk complementary and alternative medicine (Korean, Chinese, Indian, East Asian, etc.) for centuries [10,13,21]; however, the information about the mechanisms of their hair loss preventive and curative effects was lacking until recently. As a general mechanism, many herbs, higher plants, and their active constituents used for the management of alopecia target the hair cycle - keratinocyte proliferation, apoptosis, angiogenesis, microcirculation around hair bulb, hormones, and inflammation [21]. Numerous secondary metabolite-enriched extracts of plants/algae, such as *Serenoa repens* [22,23], *Panax ginseng* Mayer [24], bergamot [25], *Pinus thunbergii* [26], *Crataegus pinnatifida* [27], *Aconiti Ciliare* Tuber [28], *Caesalpinia sappan* L. [29], *Ecklonia cava* [30], and many others have been investigated in preclinical studies for their hair growth-promoting or hair loss-inhibiting actions. Among the biologically active substances, plant polyphenols, alkaloids, terpenoids have been identified, along with lipids and fatty acids, reviewed in [31,32].

A series of publications was focused on the preclinical studies of tricogenicity, an intense hair-inductive activity, of an Annurca (*Malus pumila* Miller) apple-based dietary supplement targeting follicular keratinocytes and dermal papilla cells, as well as inducing the production of trichokeratins [17,33,34,35]. The Annurca apple extracts, enriched in oligomeric procyanidins, especially procyanidin B2, have shown direct antioxidant properties in the cellular cultures, which have been considered a basis for their positive hair condition improving effects [33,35]. In animal experiments, it was noticed that the apple extract improved hair growth and conditions in rats under taxane treatment (a model of chemotherapy-induced total diffuse alopecia), due to a metabolic shift in ATP synthesis from classical aerobic glycolysis, glutaminolysis, and the Pentose Phosphate Pathway [36] to mitochondrial β-oxidation [17]. As a result, amino acids essential for keratin synthesis were spared from oxidation.

The application of high-throughput proteomic and genomic approaches to 20 vital medicinal plants from the Korean Pharmacopeia has shown that they could exert therapeutic effects on alopecia mainly by regulating phase II metabolic enzymes, such as acetylcholinesterase, phospholipase A2, ecto-5-nucleotidase, nicotinamide N-methyltransferase, and quinolinate phosphoribosyltransferase [37].

Epigallocatechin gallate in combination with rapamycin, delivered transdermally in the form of dissolvable microneedle, promoted fast hair regeneration in mice [38].

Caffeine, a hydrophilic alkaloid with high bioactivity and capacity to penetrate the lipid skin barrier, has recently attracted increased attention as an active cosmetic agent possessing antioxidant, anticellulite, antiphotoaging, and hair growth-stimulating properties [31,39]. As mechanisms of action, the improvement of local microcirculation in the scalp skin and the inhibition of the 5-α-reductase, an enzyme converting testosterone into dihydrotestosterone, which suppresses the hair shaft elongation, were hypothesized and then proven at molecular, cellular, and in vitro hair organ culture levels [40,41]. The conclusion was that enhanced hair shaft elongation, prolonged the anagen phase of the hair cycle, and stimulated hair matrix keratinocyte proliferation. However, the clinical efficacy of a topical composition of caffein with minoxidil and finasteride was moderate, and the patient’s compliance rather low [42].

An emerging trend to the prevention and treatment of hair loss is represented by the development and application of cosmeceutical and nutraceutical compositions with probiotic and prebiotic capacities. The rationale for such an approach relies on the hypothesis that defects in the communication axes, for example, the brain–gut–skin chain, could be involved in alopecia [43]. In accord with the hypothesis, this chain is employed in response to stress challenge, along in which neurogenic skin inflammation and hair growth inhibition are mediated. Therefore, the oral administration of probiotic *Lactobacillus* strain in mice dampened the stress-induced neurogenic skin inflammation associated with hair growth inhibition. Fermented preparations of *Aloe vera* applied to rats significantly promoted burn injury healing and hair growth [44]. The same effects were obtained with fermented kimchi [45]. Subsequently, this treatment improved the skin microbiota pattern, where the content of probiotic (saprophytic) bacteria such as *Lactobacillus* was increased but pathogenic bacteria (*Prevotella* and *Propionibacterium acne*) was reduced. The probiotic bacteria of the species *Leuconostoc holzapfelii* isolated from human scalp tissue, and the extracellular vesicles derived from them, were able to stimulate different hair growth-connected mechanisms [46].

Microbial (bacterial-, yeast-, fungi-, and mold-driven) fermentation has been traditionally used for the preservation of foods and the improvement of food flavor [47]. Recently, fermentation has drawn growing attention due to remarkable health potential of fermented dairy products, food supplements, and topically applied dermatological/cosmetological remedies. Fermented tropical fruits, such as papaya, noni, mangosteen, pineapple, berries, etc., seem to be a unique source of both plant-derived low molecular weight biomolecules (secondary metabolites) and fermenting microbe-associated actives with pro- and prebiotic properties [47,48,49]. Plant-derived secondary metabolites (polyphenols/terpenes/alkaloids) and microbial exometabolites/membrane components of fermented tropical fruits are known as highly bioavailable biomolecules causing skin and hair improving effects—antioxidant, wound healing, anti-inflammatory antidiabetic, antiacne, skin/hair microbiota balancing, hair growth-promoting, and hair loss-inhibiting [50,51,52,53,54]. The molecular/cellular mechanisms of these health effects have been intensely studied in in vitro and in vivo preclinical experiments. At the same time, there are a few reliable clinical data so far.

The present clinical and laboratory study has been designed and performed in order to check the effects, if any, of two cosmetic preparations (hair lotion and shampoo) containing standard preparations of fermented papaya (FP) and fermented mangosteen (FM) towards hair loss and hair quality in persons with diagnosed androgenetic alopecia and diffuse hair loss. For this purpose, a randomized double-blind placebo- and caffein-controlled clinical trial was carried out. The clinical data were assessed by questionnaires developed for the professional dermatologists–trichologists, supported by quantitative trichomicroscopy and biochemical analyses of ATP content in the hair follicles, SH-group content and protein release from the hair shaft, malonyl dialdehyde (MDA) content in the scalp skin lipids, and microbiota spectrum on the scalp skin.

## 2. Materials and Methods

### 2.1. Patients and Study Design

The study protocol was scrutinized and approved by the local ethics committee (EC of Berbekov’s State Medical University, Nal’chik, Russia; 16 December 2021 protocol). The recruitment and clinical research were carried out in two centers—Dermatology Department, Berbekov’s State Medical University and in the Center for Innovative Biotechnological Investigations NANOLAB (CIBI-NANOLAB, Trichology unit). Informed consent was obtained from all participants prior to their enrollment and data collection. Collected data included: personal and anamnestic records, trichomicrophotographs of the scalp, scalp skin sebum, and plucked hairs. Recruitment, treatment, biological material collection, and the use of personal data were conducted in accord with the Helsinki Declaration on ethics in human experimentations.

The tested cohort consisted of 154 adults (of both sexes aged from 21 to 71 years), who were clinically diagnosed as having either diffuse (*n* = 27) or androgenic alopecia (*n* = 127). They were recruited in a multicenter placebo- and competitor (caffein)-controlled clinical study (after their informed consent). All participants eligible for the efficacy study were duly informed about the purpose of the study, the manner of its conduct, and the possible side effects. Before entering the study, all of them signed an informed consent.

The control cohort consisted of 27 adults of both sexes recruited from the laboratory staff and the University students, who kindly agreed to donate plucked hairs for biochemical tests.

Eligibility criteria: (a) *Inclusion criteria*: informed consent signed; either sex; older than 18 years; (b) *Exclusion criteria*: individual hypersensitivity to any component of the cosmetic products; history of allergic reactions to cosmetic products; severe pathologies such as tumours, system blood diseases, cachexia, hypertonia of the III grade, decompensated cardiovascular states; acute and chronic diseases in the acute phase; infections of the hair-covered scalp skin.

Study design: All participants were randomly placed into 3 groups: Group 1 experimental (*n* = 100; 42 females and 58 males; age range—21–53 years; mean age—39.65 ± 2.07 years; androgenetic alopecia *n* = 83 and diffuse hair loss *n* = 17); Group 2 control placebo (*n* = 29; 17 females and 12 males; age range 26–71 years; mean age 48.58 ± 2.88 years; androgenetic alopecia *n* = 20 and diffuse hair loss *n* = 9), and Group 3 control caffein (*n* = 25; 9 females and 16 males; age range 26–62 years; mean age 43.52 ± 2.22 years; androgenetic alopecia *n* = 19 and diffuse hair loss *n* = 6). The recruited subjects were given the products for investigation free of charge in adequate quantity for 3 months of use. They were recommended to apply the lotion on the scalp daily once a day. Hair washing with shampoo was recommended 3 times a week for 3 months (14 weeks). The participants were obliged to: ➢Regular use of the products for 14 weeks;➢Avoid any other product with similar intended use;➢In the case of any side effects, they were instructed to immediately discontinue the products and consult a dermatologist.

Visits and analyses: There were 2 visits to the principal investigators: the first visit (0 weeks) was at the beginning of the study, when a person received the products and was instructed about their use. The trichomicrophotographs were taken for the first time. Several hairs from different parts of the scalp (frontal, occipital, and two temporal areas) were plucked and a scalp skin sebum sample was collected from the same scalp areas and stored for the further analyses. The second visit (14 weeks) was at the cessation of the trial, when the dermatologists/trichologists expressed their independent opinion on the treatment efficacy by filling in the questionnaire, microphotography of the scalp was taken for the second time, several hairs were plucked, and the scalp sebum was collected as described above.

### 2.2. Questionnaire for Experts Dermatologists/Trichologists

The questionnaire containing dermatologists’/trichologists’ (*n* = 4) professional independent opinions on the treatment efficacy was filled in at the beginning and at the cessation of the trial. The answers were scored depending on the initial state of hair, and the degree of the effect at trial cessation: 0 (negative effect), 1 (no effect), 2 (moderate effect), 3 (good effect), 4 (very good effect), and 5 (excellent effect). The average scores for the groups and for the time of answers were calculated. The data were presented as the mean ± SD.

### 2.3. Cosmetic Products under Investigation and Protocols of the Application

Standardized fermented tropical fruits papaya (*Carica papaya* L.) and mangosteen (*Garcinia mangostana* Linn.) syrups of food grade [50,51,52,53,54] were used as active ingredients for hair care cosmetic products (they are classified as raw materials for cosmetics by the International Cosmetic Ingredient Index (INCI)). Their concentrations in the experimental lotion were equal to 2.0% *w*/*w* for each fermented fruit, and caffeine concentration was 0.5% *w*/*w* (lotion’s internal code, MEDENA-1324-HL). In the experimental shampoo, contents of fermented fruits were 0.5% *w*/*w* each, and caffeine was absent (shampoos’ internal code, MEDENA-1348-HS). The placebo preparations of lotion/shampoo did not contain any active ingredient, but only standard excipients used as a basis for hair lotions and shampoos manufactured by the company. The caffeine control lotion contained exclusively 0.5% *w*/*w* caffeine as an active ingredient.

### 2.4. Instrumental Method of Hair Loss and Hair Quality Assessment

The instrumental method of the evaluation was a quantitative video-registration performed by an ARAMO SG videocamera connected to a computer equipped with the Trichoscience ver. 1.6 program. At visits 1 and 2, the dermatologist–trichologist doctor thoroughly inspected the occipital and frontal scalp areas by a computer-connected device with laser-optic-equipped microscope lenses (magnification ×10, ×100, and ×200) and TV camera. Microphotographs and calculations were implemented to determine the following parameters: hair density (hair number per 1 cm^2^), average hair thickness (diameter, μm), per cent (%) of thin hairs (diameter from 30 to 50 μm), per cent (%) of thick hairs (diameter > 70 μm), per cent (%) of medium hairs (diameter from 50 to 70 μm), root state, and hair shaft conditions. To determine hair density, lenses ×60 were used (Figure 1).

To assess average hair thickness, diameter of hairs (μm) was determined using a ×200 lens (Figure 2). The numbers in yellow squares indicate the hair diameter in μm.

The follicle state (diameter in μm) and hair shaft (fat and water presence) conditions were quantified as well (Figure 3).

### 2.5. Lipid Peroxidation Assay

Skin surface and hair lipids were collected from 3 points of the scalp by application of cotton disks (diameter 5.3 cm) soaked in diethyl ether, allowing contact for 2 min. Then, the freshly collected lipids were pooled, filtered, and evaporated. The content of malonyl dialdehyde (MDA) in the scalp skin and hair lipids was measured by a quantitative spectrophotometric method using OxiSelect TBARS Assay Kit (Cell Biolabs, Inc., MA, USA). The data were expressed in μmol/cm^2^.

### 2.6. ATP Assay in Hair Follicles

During visits 1 and 2, the dermatologist–trichologist doctor plucked a few hairs from occipital, frontal, and temporal scalp areas to analyze energy (ATP) content in a single hair follicle. In brief, plucked hairs (Figure 4a) were cut into 5 mm pieces, delivered to biochemical laboratory on ice, and processed immediately. Five single hair pieces proximal to the hair bulb were selected and dissected into smaller pieces under a microscope (Figure 4b). The dissected follicles were placed into 200 μL potassium phosphate buffer (0.1 M, pH 7.8) supplied with fetal calf serum at room temperature for 60 min. Then, the samples were shaken vigorously for 60 s on a Vortex. The liquid parts of samples were collected and the content of adenosine-3phosphate (ATP) was determined by the luciferin-luciferase method using a microfluorimeter (Hitachi, JOsaka, apan).

The principle of ATP assay is based on the quantitative bioluminescent determination of adenosine 5’-12 triphosphate (ATP), assessed by the Bioluminescence Assay Kit. In the assay, ATP is consumed when firefly luciferase catalyses the oxidation of D-luciferin to adenyl-luciferin which, in the presence of oxygen, is converted to oxyluciferin with light emission. This latter reaction is essentially irreversible. When ATP is the limiting reagent, the light emitted is proportional to the ATP present. The measurements of luciferin-luciferase chemiluminescence were performed on a Victor 2 1420 multilabel counter, equipped with Wallac 1420 Software (Perkin Elmer, Waltham, MA, USA). The quantity of ATP in hair bulbs was determined using a calibration curve with definite amounts of standard ATP solutions. Results were expressed in nmol ATP/5 follicles [55].

### 2.7. Determination of SH-Group Content and Protein Leakage from Hair Shaft

The distal pieces of cut hairs were used to analyze the content of thiol groups (SH-groups) in small labile proteins leaking from the hair shaft, as well as the content of leaking proteins themselves. In brief, hair pieces were weighted and 2–3 mg of hairs was placed into 0.75 mL 0.1 M potassium phosphate buffer (pH 7.4), destroyed in a glass homogenizer, and centrifuged to sediment the particles of the destroyed hairs. The supernatants were analyzed for protein and SH-group contents. The supernatants (0.35 mL) were mixed with TRIS buffer (0.65 mL, pH 8.9) and 25 μL of DTNB solution. Then, SH-groups were determined with the HPLC technique on a HPLC Shimadzu SCL-10AV (Shimadzu Europa GmbH, Duisburg, Germany) chromatograph equipped with array photodiode and electrochemical detection, as described previously [56]. Namely, quantitative analysis of total SH-groups was carried out on an analytical Supercosil LC-NH_2_ column (25 cm, 4.6 mm, 5 mm) with the following parameters: detection, 355 nm; mobile phase, A 1/4 MeOH/H_2_O (65/35, *v*/*v*), B 1/4 X/Y (20/80, *v*/*v*), X (sodium acetate 3H_2_O, 272 g; acetic acid, 372 mL; H_2_O, 122 mL), Y (MeOH/H_2_O) (80/20, *v*/*v*); gradient program, %B 1/4 5 for 15 min, %B 1/4 90 in 30 min, and %B 1/4 90 in 45 min; flow, 1 mL/min). The results were expressed in ng/mg protein.

The protein content was measured by the Bradford method [57] using a microplate assay kit (Bio-Rad, Hercules, CA, USA).

### 2.8. Scalp Skin Microbiota Determination

The total amount of facultative anaerobic microbes on the scalp skin was determined by routine microbiological test of sectoral seeding, as described elsewhere [53]. Skin swabs were taken by cotton disks (diameter 5.3 cm) soaked in the sterile physiological solution. Two or three points of the frontal and occipital scalp skin were determined and used before and after the treatment. The isolation and identification of bacteria was performed using microbiological processes followed by microscopic observation and confirmed by mass spectrometry MALDI-TOF (Microflex, Brucker, USA) [58,59].

### 2.9. Statistical Analysis

The statistical analysis of the clinical data was carried out using WINSTAT programs for personal computers (Statistics for Windows 2007, Microsoft, MA, USA). All biochemical and microbiological measurements were performed in triplicate, and the data were statistically evaluated. The reported values were treated as continuous. The normality of data was confirmed using the Shapiro–Wilk test. Since the distribution of the data was significantly different from normal, nonparametric statistics were used. The results were expressed as the mean ± SD. Mann–Whitney U test was employed for comparison between independent groups of data. To evaluate the difference between connected data, the two-tailed Student’s *t*-test was applied and *p* < 0.05 was considered significant. Fisher’s exact test was used to determine the significance of the differences between the beginning and cessation of the trial while assessing the scores of subjective opinions of patients on the effects of cosmetics. For these tests, *p* < 0.05 was considered significant and 0.05 < *p* < 0.1 indicated a trend toward significance.

## 3. Results

### 3.1. Assessment of Clinical Efficacy of the Hair Care Cosmetics

All recruited persons successfully completed the study. There were no dropoffs due to adverse effects or low patients’ compliance to the treatment. Mean values of the scores assigned by four independent dermatologists/trichologists who filled in the questionnaire at the beginning and cessation of the trial are collected in Table 1. Of importance, the experimental treatment (Group 1) led to clear-cut improvement in hair shaft conditions, fat quantity, and hair loss intensity. For control Groups 2 and 3, diminished fatness of hair/scalp was the only feature that has been changed statistically significantly.

### 3.2. Instrumental Assessment of Hair Loss Prevention and Hair Quality Improvement Effects of the Hair Care Cosmetics

Microscopic analysis of hair diameter (μm), hair density (hair number/cm^2^), and the percentage of thick hairs (Figure 5, Figure 6 and Figure 7, respectively) in the frontal and/or occipital areas clearly showed positive effects of the experimental treatment for 60 days, and the absence of statistically significant changes (in red shown in (a) plates) in both control groups. Determination of the presence of medium and thin hairs (Figure 8 and Figure 9, respectively) resulted in increased numbers of medium hairs for both control groups, while significant decrease in thin hairs was observed in all groups.

Microphotographs are available to confirm the hair density changes, if any, in the patients of Groups 1, 2, and 3 (see Appendix A, Figure A1, Figure A2 and Figure A3, respectively).

### 3.3. Effects of the Hair Care Products on the Intensity of Lipid Peroxidation in the Hair and Scalp Skin Lipids

The intensity of lipid peroxidation assessed by the levels of malonyl dialdehyde (MDA) in the lipid part of the scalp skin hydrolipid mantel was verified in the pool of biomaterial collected from four distant areas of the scalp (see the Material and Methods subsection). The results are presented in Table 2. It is seen that regular application of the hair care products for 60 days in accord with the protocol led to remarkable inhibition of free radical-driven lipid peroxidation exclusively in the experimental group (**Group 1**), while in placebo group (**Group 2**) and caffein group (**Group 3**) the intensity of lipid peroxidation remained unchanged. The content of the end-product of lipid peroxidation in the scalp skin and hair lipids of the experimental group of alopecia patients practically reached the normality range. The range of MDA content in healthy donors (*n* = 27) without complaints to hair loss was 18.3 ± 1.4 μmol/cm^2^.

### 3.4. Comparative ATP Content (nmol/5 Follicles) in the Hair Follicles of Participants with Alopecia Androgenetica and Diffuse Hair Loss and Healthy Donors

In the beginning of the trial, ATP levels in hair follicles taken from the frontal and occipital areas of healthy donors were significantly greater than those in patients diagnosed either with alopecia androgenetica or with diffuse hair loss (Table 3). At the same time, the ATP content was similar for both types of alopecia.

### 3.5. Effects of the Hair Care Products on the Hair Root/Follicle Diameter (μm) and Content of ATP (nmol/5 Follicles) in the Hair Follicles

The course of hair care cosmetics application according to the protocol of the clinical trial resulted in a remarkably significant increase in ATP content in hair follicles (taken from both frontal and occipital areas of the scalp) of the participants from the experimental Group 1 and the caffein control Group 3, while there were no changes in the placebo control Group 2 (Table 4). Energy storages in hair follicles of Group 1 were statistically significantly increased as compared to the initial point of the trial and compared to both control Groups 2 and 3. This increase observed for the Group 3 (caffein control) was less evident versus Group 1 (experimental). However, the difference was statistically significant comparing with the initial levels of ATP and the levels in placebo control Group 2. Regarding hair root/follicle diameter that corresponds to the root state (the larger diameter due to swelling of the root, the worse are the root conditions), the data are collected in Table 4. Hair root conditions were significantly improved exclusively in Group 1 (experimental) (*p* < 0.05 versus the initial point).

### 3.6. Effects of the Hair Care Products on the Protein Leakage from the Hair Shaft and the SH-Group Content in the Hair Shaft

The leakage of labile, mainly low molecular weight proteins through damaged hair shaft cuticles was rather low in both frontal and occipital areas of the scalp of healthy donors (Table 5). This parameter was increased substantially in both groups of participants diagnosed with alopecia. At the same time, there was no difference between the two alopecia groups.

When the background ATP content in hair follicles of the patients with alopecia (androgenetic and diffuse hair loss) was plotted against the labile protein leakage from the hair shaft in order to reveal a relationship—if any—between the values (Figure 10), no correlation was found. It seems that these markers were independent (see the red line of correlation). However, several scattered individual data showed that the greater protein leakage corresponded to the lower range of ATP content, while the lower protein leakage was characteristic for the follicles with high ATP content.

The dynamics of labile protein leakage for the three groups during the clinical trial are shown in Table 6. At the initial point of the study, average values were significantly higher than normal and did not differ among the groups. The treatments with experimental or control hair care products resulted in diminished protein leakage. These post-trial values were statistically indistinguishable for all three groups. When non-binding SH-groups were measured for the hair shaft at the beginning of the trial, their average content was expectedly higher than normal (the normality range is between 7 and 15 ng/mg protein) in all patients with androgenetic alopecia or diffuse hair loss (Table 6). This value was practically normalized in the experimental group by the trial cessation, dropped significantly in the caffein-control group, and remained at the initial level in the placebo-control group.

### 3.7. Effects of the Experimental Hair Care Products on the Microbiota Spectrum on the Scalp Skin

Twenty five strains of microorganisms usually present on the surface of human skin were analyzed. These microorganisms belong to the following groups: cocci, bacilli, facultative anaerobic microbes, actinobacteria, and yeast-like fungi. Most microorganisms analyzed were absent and did not appear after the clinical study cessation. Among them were cocci/bacilli (*Bacillus cereus*, *Bacillus megaterium, Enterococcus* spp., and *Staphylococcus aureus*) and facultative anaerobic bacteria (*Bacteroides fragilis*, *Bifidobacterium* spp., *Blautiacoccoides*, *Clostridium* spp., *Clostridium difficile*, Cl. *hystoluticum/Str.pneumonia*, *Clostridium perfringens*, *Fusobacterium spp.*/*Haemophilus* spp., *Peptostreptococcus anaerobius 18623*, *Peptostreptococcus anaerobius 17642*, *Prevotella* spp., *Ruminicoccus* spp., and *Veilonella* spp.).

At the same time, the presence of several pathogenic microorganisms, such as *Streptococcus* spp., *Staphylococcus epidermidis*, *Propionibacterium acnes*, *Cutibacterium acne*, and *Malassezia restructa*, was significantly restricted after the trial (Table 7), while the presence of *Actinomyces* spp., *Corynebacterium* spp., and *Malassezia globosa* remained unchanged and within the normal range of the values.

## 4. Discussion

In the present clinical–laboratory study, we attempted for the first time to show the clinical efficacy of cosmetic hair care preparations containing the well-established and extensively studied food supplements—fermented papaya and mangosteen—as active agents against androgenetic alopecia and diffuse hair loss. The primary goal of this randomized double-control (placebo and caffein) clinical investigation was to quantify clinical efficacy. The secondary goal was to identify biochemical markers relevant for the efficacy observed. The professional opinion of experts in the field was collected as scores, and showed that exclusively experimental lotion and shampoo containing fermented papaya and mangosteen as well as caffein improved hair shaft conditions and attenuated hair loss (Table 1). The hair fatness was fairly diminished in all three groups, experimental and controls alike. Very likely, excipients of the cosmetic compositions, which were similar for all products in question, effectively eliminated an excess of hair and scalp fat. The instrumental assessment of clinical efficacy gave clear-cut results, that the experimental products remarkably increased hair density in different scalp areas, median hair shaft diameter, and a percentage of thick hairs (Figure 5, Figure 6 and Figure 7). Simultaneously, thin hair percent was significantly decreased (Figure 8). In both control groups, the quantity of thin hairs was diminished as well, while the medium diameter hairs were increased (Figure 8 and Figure 9). We could interpreted these sets of data as evidence that the experimental compositions with fermented tropical fruits either prevented hair loss or induced hair growth; therefore, the density of hair was ameliorated. We are inclined to explain the observed effects as hair loss prevention, because the pathological pattern of hair roots (swelling assessed by the root diameter) in the beginning of the trial was substantially improved after application of the experimental hair care cosmetics (Table 4). These compositions were also effective toward hair shaft conditions by preventing hair thinning and by shifting thin hairs mainly to the thick category. Both control cosmetics (placebo- and caffein-control) also slightly improved hair shaft conditions, shifting thin hairs to medium diameter hairs. Similar clinical effects on androgenetic hair loss have been observed with topical cosmetic compositions containing caffein, zinc, and plant polyphenols [1], or with the dietary supplement based on the Anurca apple [33].

Only a few clinical studies have been carried out on diffuse (stress- or chemotherapy-connected alopecia (reviewed in [13]). Caffein- and zinc-containing substances have been used clinically in the attempt to decrease hair fall, thinning, and shedding [60]. In our case, patients with diffuse hair loss were randomly included into experimental and control groups. On these grounds, we concluded that the experimental cosmetics with fermented papaya and mangosteen prevented hair loss and improved hair thickness in patients with diffuse hair loss caused by chemotherapy and psychological stress.

The etiology and pathogenesis of androgenetic and diffuse hair loss and the molecular mechanisms underlying these two pathologies are quite different (see references above and in the Introduction). Since similar clinical effects were observed in patients diagnosed with the two types of alopecia, we attempted to search for biochemical markers characteristic for the same. One of the possible markers was ATP content in the hair follicle/root. As seen in Table 3, energy stores dropped dramatically in the follicles of patients with both types of alopecia as compared to healthy donors. This energy deficiency practically disappeared after treatment with cosmetics containing fermented papaya and mangosteen (Table 4), and increased substantially in the caffein-control Group 3. The roles of keratinocyte-located ATP in physiology of fast-growing hair are numerous and essential—cell proliferation and movement, cellular metabolism, and keratin synthesis. Therefore, ATP deficiency or changes in its metabolism, for example, excessive formation of cyclic adenosine monophosphate (cAMP), has tremendous negative impact on hair loss and growth as well as on hair conditions, such as hair root deformation, hair thinning, and shedding. The most effective pharmacological approach for the cure of androgenetic alopecia is minoxidil, which opens ATP-dependent K-channels [12]. The ATP synthesis could be inhibited by the lack of nutrients, mitochondrial insufficiency, and alteration of scalp skin microcirculation. On the other hand, the ATP expenditure could be elevated by hyperactivated ATP-dependent metabolic and cellular processes. Thus, ATP deficiency could occur. For the time being, the molecular pathways of ATP deficiency and its role in the hair loss of different etiology remain obscure.

The background ATP deficiency in hair follicles of patients with both types of alopecia did not correlate with the leakage of low molecular weight proteins from the matrix of hair shafts (Figure 10). Moreover, protein leak characteristic for hairs damaged by environmental or artificial conditions (UV radiation, hair dyeing or chemical/thermal hair curling/waving) was equally diminished by regular application of experimental and control hair care products, which probably reflected the healing properties of basic excipients of lotions and shampoos toward damaged hair cuticles (Table 6), independently of the presence of biologically active agents with hair loss-preventing and hair thickness-inducing properties. Therefore we came to the conclusion that this biochemical marker could be omitted in the testing protocols for hair loss-preventing cosmetics.

Conversely, the measurement of the SH-groups in low molecular weight proteins/peptides containing cysteine and methionine, which form an amorphous hair matrix of high sulfur proteins, could be useful in the screening of effective hair care products [61]. These high-sulfur peptides do not participate in the formation of bonds between fibrous proteins of the hair. The content of these groups is enhanced in hair subjected to exposure to UV radiation [62]. Table 6 demonstrates that the experimental treatment led to a significant drop in reduced SH-groups in Group 1, to a moderate drop in the caffein-control Group 3, and to no effect in the placebo-control Group 2. We assume that SH-groups could be reused for the formation of S-S bonds between high-molecular weight keratins synthesized upon action of caffein, or of a combination fermented fruits/caffein in the thickening hairs. However, more research should be carried out to confirm our assumption.

Not surprisingly, experimental hair care cosmetics diminished the levels of MDA, a final product of free radical-driven lipid peroxidation of hair and scalp skin lipids (Table 2). Fermented papaya and mangosteen are recognized as excellent antioxidants, with their antioxidant capacity greatly enhanced by the fermentation process. The leading multiple roles of oxidative stress have been appreciated for all types of alopecia. The cartoon of Figure 11 collects possible cellular, chemical, and biochemical sources of free radicals in patients diagnosed with androgenetic, areata, and diffuse alopecia.

The effects of experimental hair care products on microbiota residing on the scalp skin and hair shaft (Table 7) confirmed our hypothesis that the presence of fermented papaya and mangosteen with pre- and probiotic components [53,54] could positively change the altered pattern of microbiota in cases of increased hair loss [44,45,46]. While the content of several pathogens was decreased, the saprophytes (probiotics) remained in the same quantities, and the whole spectrum of microorganisms was not depleted. One of the possible mechanisms of the probiotic activity of fermented papaya is the selective inhibition of catalase in the microbial pathogens, which employ this enzymatic antioxidant activity as a defensive mode against immune response of the host organism [53]. This microbiota-correcting effect could impact the clinical efficacy of experimental cosmetics observed in the trial. Unfortunately, the absence of microbiota measurements in the control groups (due to the high cost of the analysis) is a major limitation in the present research.

More investigations on the preclinical and clinical effects of food-grade fermented products as active components for cosmetics are needed in order to provide a solid background for their cosmetic effects. A combination of phytochemical and biological experiments would allow for identification of plant- and microbe-derived substances with specific hair loss prevention and hair quality enhancement effects. Recently, a new aliphatic ester of hydroxysalicilic acid isolated from fermented *Carica papaya* L. has been found responsible for hair growth-stimulating activity in the preparation [63].

## Figures and Tables

**Figure 1 biomolecules-13-00699-f001:**
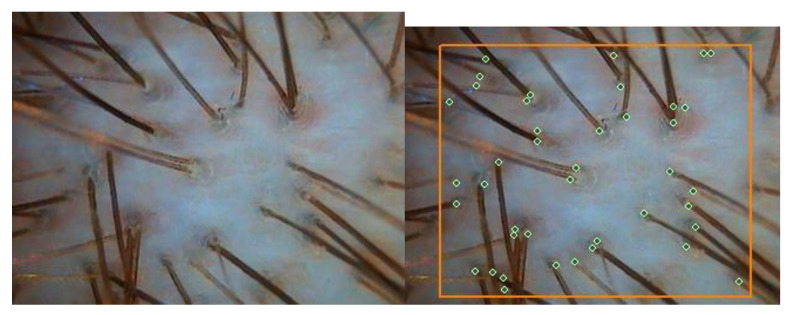
Determination of hair density. One cm^2^ is bordered by the orange line. Magnification ×60.

**Figure 2 biomolecules-13-00699-f002:**
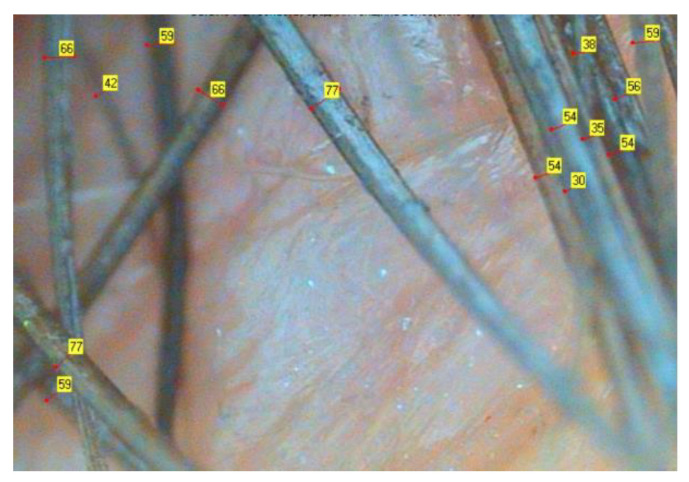
The determination of a single hair diameter was conducted under ×200 magnification lenses. The numbers in yellow squares indicate the hair diameter in μm.

**Figure 3 biomolecules-13-00699-f003:**
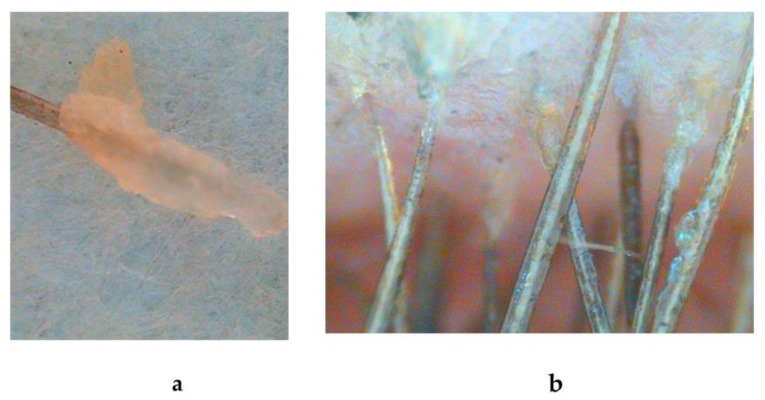
Determination of the hair follicle diameter and lipid/water mantel conditions: (**a**) hair root/follicle; (**b**) lipid mantel and hydration conditions were performed under ×200 magnification.

**Figure 4 biomolecules-13-00699-f004:**
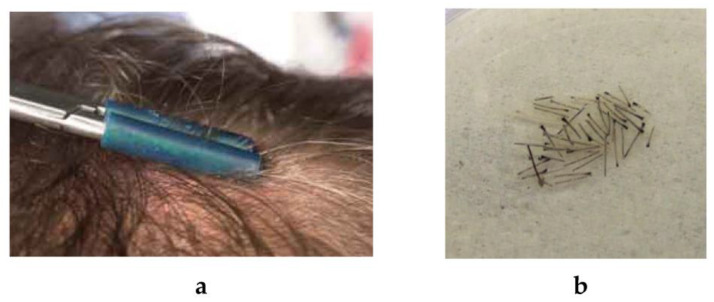
Preparation of hair follicles for the ATP measurements in the hair follicles: (**a**) hair plucking; (**b**) separated hair follicles.

**Figure 5 biomolecules-13-00699-f005:**
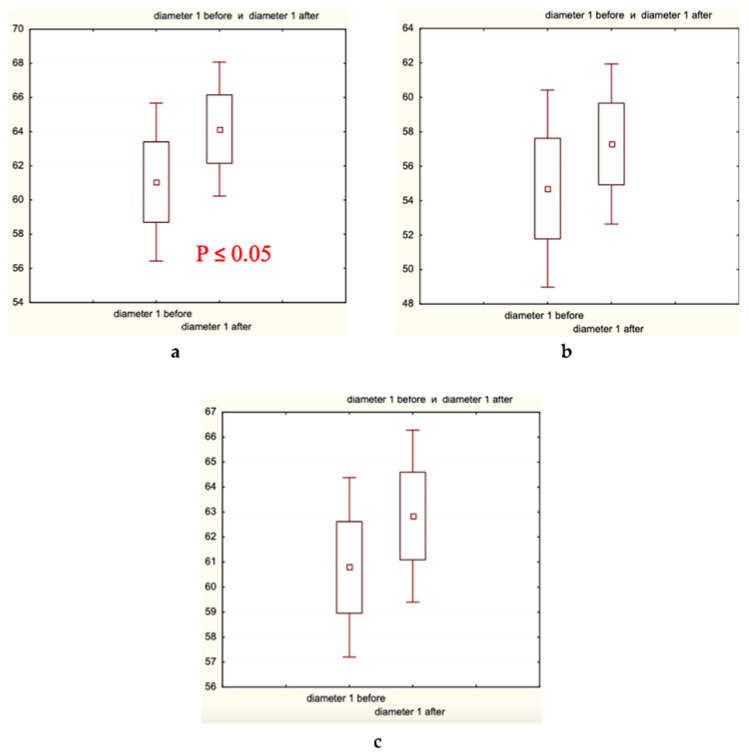
Hair diameter (μm) distribution before and after the trial: (**a**) Experimental Group 1; (**b**) Placebo Control Group 2; (**c**) Caffeine Control Group 3. Small boxes—median values; large boxes—median values ± standard error of median (SEM); whiskers—median values ± 1.96 SEM. *p* < 0.05.

**Figure 6 biomolecules-13-00699-f006:**
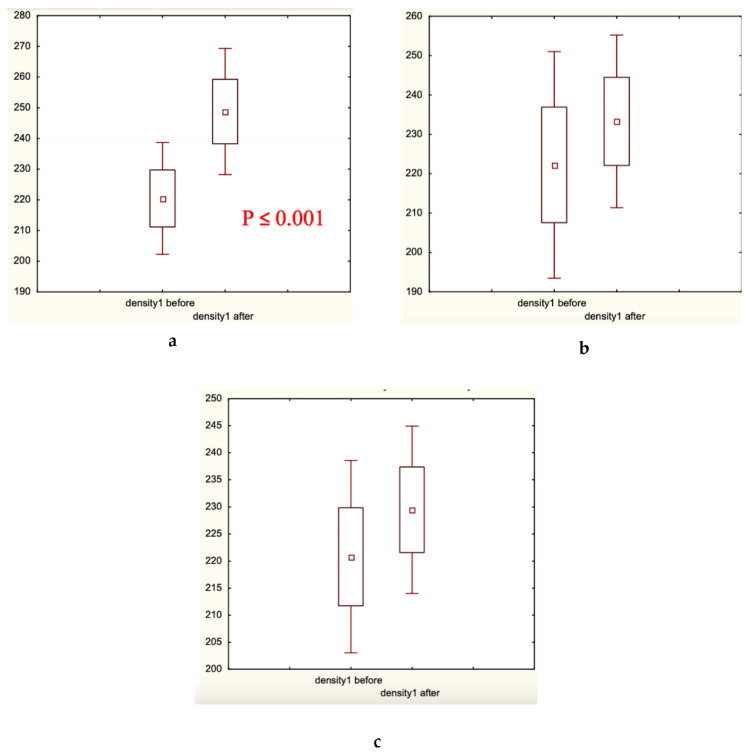
Hair density (*n*/cm^2^) distribution before and after the trial: (**a**) Experimental Group 1; (**b**) Placebo Control Group 2; (**c**) Caffeine Control Group 3. Small boxes—median values; large boxes—median values ± standard error of median (SEM); whiskers—median values ± 1.96 SEM. *p* < 0.001.

**Figure 7 biomolecules-13-00699-f007:**
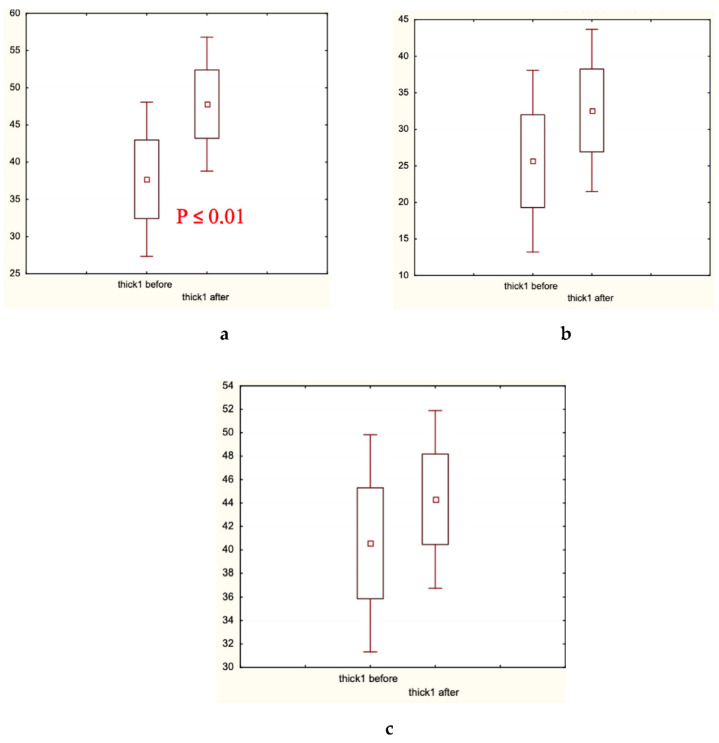
Thick hair presence (%) before and after the trial: (**a**) Experimental Group 1; (**b**) Placebo Control Group 2; (**c**) Caffeine Control Group 3. Small boxes—median values; large boxes—median values ± standard error of median (SEM); whiskers—median values ± 1.96 SEM. *p* < 0.01.

**Figure 8 biomolecules-13-00699-f008:**
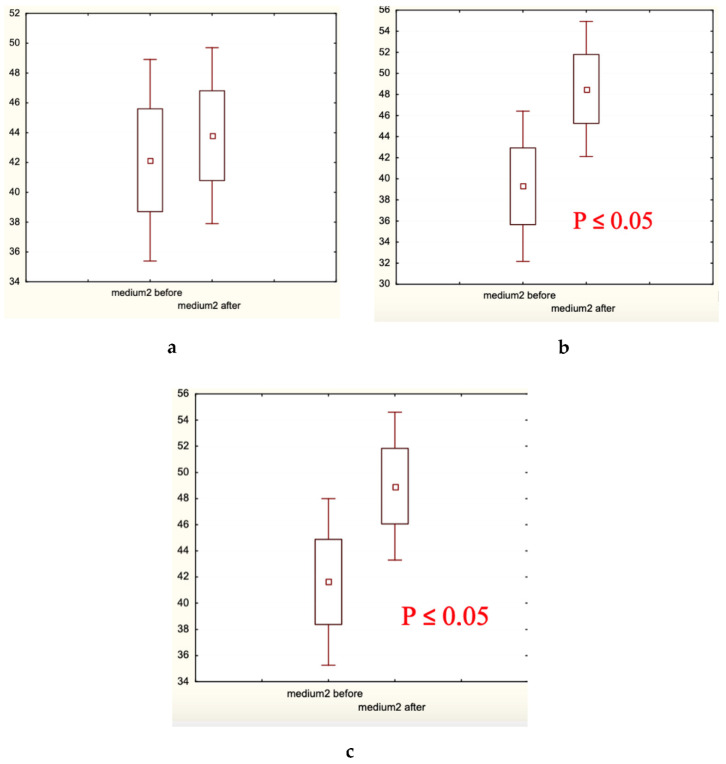
Medium hair presence (%) before and after the trial: (**a**) Experimental Group 1; (**b**) Placebo Control Group 2; (**c**) Caffeine Control Group 3. Small boxes—median values; large boxes—median values ± standard error of median (SEM); whiskers—median values ± 1.96 SEM. *p*< 0.05.

**Figure 9 biomolecules-13-00699-f009:**
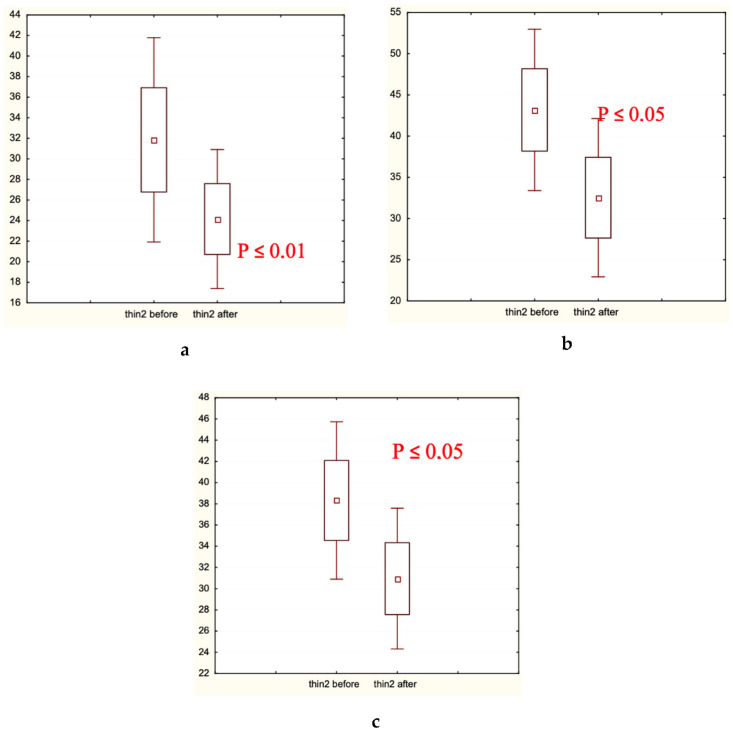
Thin hair presence (%) before and after the trial: (**a**) Experimental Group 1; (**b**) Placebo Control Group 2; (**c**) Caffeine Control Group 3. Small boxes—median values; large boxes—median values ± standard error of median (SEM); whiskers—median values ± 1.96 SEM. *p* < 0.05, *p* < 0.01.

**Figure 10 biomolecules-13-00699-f010:**
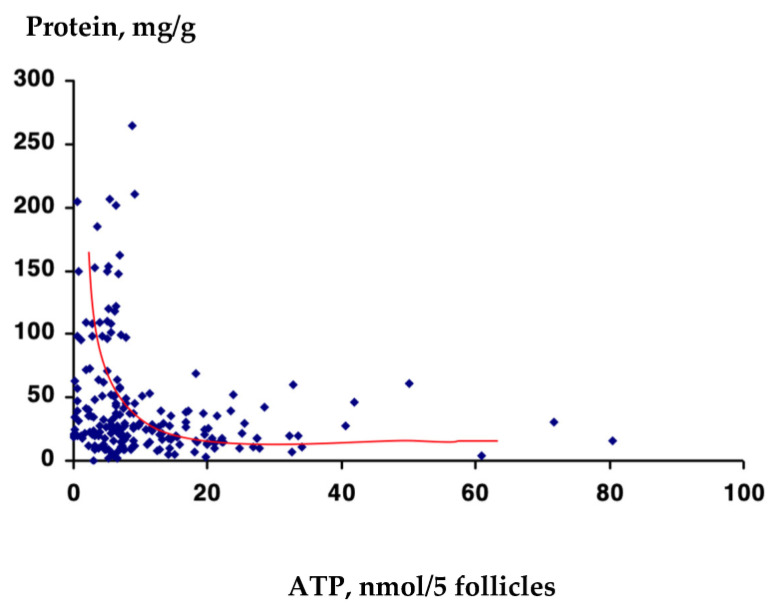
Relationship between ATP content in hair follicles and labile protein leakage from the hair shaft in patients with two types of alopecia (*n* = 154). Blue dots represent values for the individual person. Red line is a correlation curve.

**Figure 11 biomolecules-13-00699-f011:**
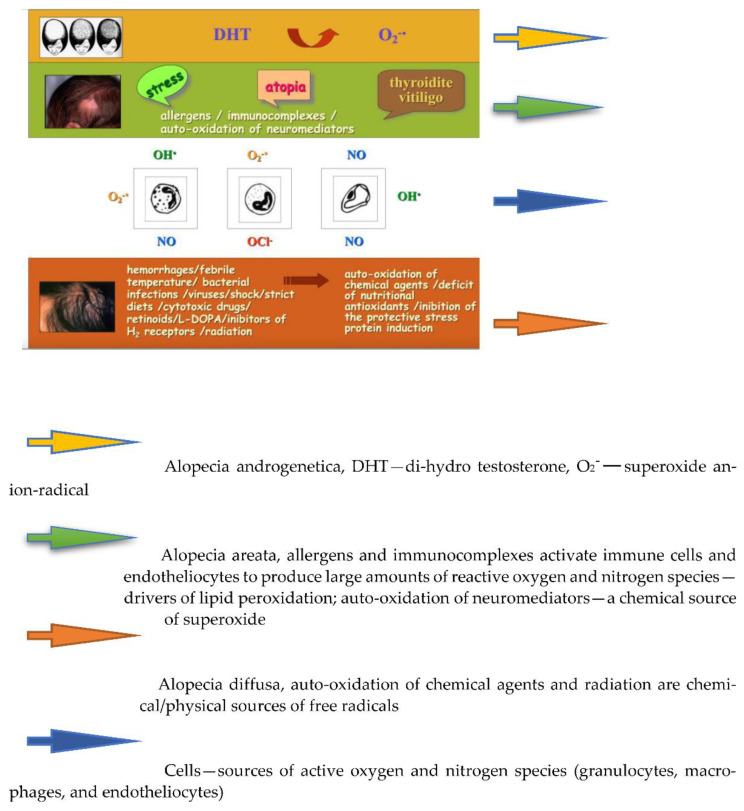
Sources of oxidative stress in different types of alopecia.

**Table 1 biomolecules-13-00699-t001:** Dynamics of subjective professional clinical assessment (score) by a dermatologist/trichologist for the hair conditions in the three groups of participants (*n* = 154) before and after the trial.

Symptom	Group 1 (*n* = 100)		Group 2 (*n* = 29)		Group 3 (*n* = 25)	
	Before	After	Before	After	Before	After
**Hair shaft conditions**	3.0 ± 0.5	4.3 ± 0.3 **	3.1 ± 0.4	3.3 ± 0.7	3.1 ± 0.5	3.4 ± 0.5
**Scalp skin dryness**	1.7 ± 0.2	1.6 ± 0.1	1.6 ± 0.2	1.5 ± 0.2	1.5 ± 0.3	1.7 ± 0.2
**Scalp skin redness**	1.2 ± 0.1	1.0 ± 0.2	1.0 ± 0.1	1.0 ± 0.1	1.2 ± 0.1	1.0 ± 0.0
**Scalp skin/hair fatness**	1.8 ± 0.2	1.0 ± 0.1 **	2.0 ± 0.3	1.4 ± 0.1 *	2.1 ± 0.2	1.4 ± 0.1 **
**Scalp skin irritation**	1.2 ± 0.1	1.0 ± 0.0	1.0 ± 0.1	1.0 ± 0.1	1.0 ± 0.2	1.0 ± 0.0
**Hair loss intensity**	2.4 ± 0.2	1.8 ± 0.2 **	2.5 ± 0.3	2.3 ± 0.2	2.2 ± 0.2	2.1 ± 0.3

*, **—before versus after *p* < 0.05 and *p* < 0.01, respectively. Group 1 participants were treated by hair care cosmetics (shampoo and lotion) containing fermented papaya and mangosteen, and caffein as actives for 14 weeks; Group 2 subjects were treated by placebo shampoo and lotion for 14 weeks; Group 3 participants were treated by lotion containing caffeine and placebo shampoo for 14 weeks.

**Table 2 biomolecules-13-00699-t002:** Effects of hair care product application on lipid peroxidation (MDA) in the scalp skin and hair lipids.

Group	Content of MDA, μmol/cm^2^	
	Before	After 14 Weeks
**Group 1, Experimental (lotion/shampoo containing fermented fruits and caffein)**	23.8 ± 1.6	20.0 ± 1.5 *
**Group 2, Placebo Control (lotion/shampoo without actives)**	24.0 ± 1.3	23.9 ± 2.1
**Group 3, Caffeine Control (lotion/shampoo with caffein only)**	24.2 ± 1.2	24.5 ± 1.5

*p* < 0.05 before versus after. Normal range of values is 18.3 ± 1.4 μmol/cm^2^.

**Table 3 biomolecules-13-00699-t003:** Energy stores in hair roots/follicles taken from the frontal and occipital areas of healthy donors and patients with different types of alopecia (ATP content, nmol/5 follicles).

Group	ATP Content (nmol/5 Follicles)	
	Frontal Area	Occipital Area
**Healthy donors**	80.6 ± 9.1 **	74.8 ± 7.4 **
**Alopecia androgenetica**	19.9 ± 3.2	17.8 ± 2.7
**Diffuse hair loss**	24.8 ± 4.0	21.1 ± 1.8

**—*p* < 0.01 versus both alopecia groups.

**Table 4 biomolecules-13-00699-t004:** Dynamics of hair follicle diameter and ATP content in the follicles.

Group	Hair Root/Follicle Diameter (μm)		ATP Content (nmol/5 Follicles)	
	Before	After	Before	After
**Group 1** Experimental (lotion/shampoo containing fermented fruits and caffein)	98 ± 13	74 ± 12 *	20.5 ± 2.9	71.5 ± 3.1 **^, #^
**Group 2** Placebo Control (lotion/shampoo without actives)	96 ± 15	92 ± 11	17.7 ± 3.2	18.9 ± 2.8
**Group 3** Caffeine Control (lotion with caffein/placebo shampoo)	97 ± 10	104 ± 16	18.5 ± 3.0	40.2 ± 4.8 *^, #^

*—*p* < 0.01 versus the initial point; **—*p* < 0.001 versus the initial point; ^#^—*p* < 0.01 versus control group(s).

**Table 5 biomolecules-13-00699-t005:** Labile protein leakage (μg/g hair) from hair shafts taken from the frontal and occipital areas of healthy donors and patients with different types of alopecia.

Group	Labile Protein Leakage from Hair Shaft (μg/g Hair)	
	Frontal Area	Occipital Area
**Healthy donors**	15.0 ± 0.9 **	15.3 ± 1.2 **
**Alopecia androgenetica**	49.2 ± 9.8	48.5 ± 7.0
**Diffuse hair loss**	47.9 ± 8.8	48.1 ± 9.6

**—*p* < 0.01 versus alopecia groups.

**Table 6 biomolecules-13-00699-t006:** Dynamics of labile protein leakage (μg/g) from the hair shaft and non-binding SH-groups (ng/mg protein) in the hair shaft.

Group	Labile Protein Leakage from Hair Shaft (μg/g Hair)		Nonbinding SH-Groups in Hair Shaft (ng/mg Protein)	
	Before	After	Before	After
**Group 1** Experimental (lotion/shampoo containing fermented fruits and caffein)	46.5 ± 7.2	21.3 ± 4.5 **	35.7 ± 5.0	12.3 ± 2.5 **^, #^
**Group 2** Placebo Control (lotion/shampoo without actives)	47.1 ± 9.0	28.9 ± 5.1 *	41.1 ± 6.7	53.7 ± 7.6
**Group 3** Caffeine Control (lotion with caffein/placebo shampoo)	45.9 ± 8.4	20.8 ± 4.4 **	33.2 ± 2.5	24.1 ± 1.9 *^, #^

*—*p* < 0.05 versus the initial point; **—*p* < 0.01 versus the initial point; ^#^—*p* < 0.01 versus control group(s).

**Table 7 biomolecules-13-00699-t007:** Microbiota-correcting effects of the experimental hair care cosmetics.

Microorganisms	Normal Values, 10^5^ Cells/g	Clinical Trial Data, 10^5^ Cells/g		*p* between before and after
		Before	After	
*Streptococcus* spp.	270–350	355 ± 15	300 ± 15	*p* < 0.05
*Staphylococcus epidermidis*	180–240	235 ± 15	190 ± 15	*p* < 0.05
*Propionibacterium acnes*	0–42	35 ± 5	25 ± 3	*p* < 0.05
*Cutibacterium acne*	0–50	55 ± 10	35 ± 4	*p* < 0.05
*Malassezia restructa*	0–42	25 ± 5	10 ± 3	*p* < 0.01
*Actinomyces* spp.	52–92	75 ± 10	77 ± 20	*p* > 0.05
*Corynebacterium* spp.	575–635	600 ± 50	590 ± 30	*p* > 0.05
*Malassezia globosa*	0–52	20 ± 5	25 ± 3	*p* > 0.05

## Data Availability

Not applicable.

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
