# Peer review of "Biomolecules of Fermented Tropical Fruits and Fermenting Microbes as Regulators of Human Hair Loss, Hair Quality, and Scalp Microbiota"

_biomolecules, 2023, doi:10.3390/biom13040699_

Round 1

Reviewer 1 Report

Hair loss is not a critical issue on human health, but many people have an interest about preventing hair loss and improving hair growth with different applications. 

Authors recruited human patients and performed an experiments for 14 weeks which is very time and effort consuming work. I have no doubt about the analysis of indicated factors and scoring. 

However, as clinical trials of human treatment there should be a picture before/after treatment. As authors mentioned, caffeine containing fermented fruit shampoo effect should be demonstrated with picture if the treatment is efficiently working on the hair growth. 

Please, add figures of group 1, 2 and 3 before and after treatment. 

Author Response

Answer: As per your advice, we added the Annex containing 3 Figures (Figure 1A, 2A, and 3A) showing the examples of hair density micro-photos taken before and after the clinical trial for all 3 Groups (experimental, placebo control, and caffein control). The Annex is placed after the References sub-chapter. We keep this kind of micro-photographs for ALL 154 participants of the trial in our Institutional on-line archives. The link could be provided upon request.

Reviewer 2 Report

Comments to authors:

1.      Authors could add graphical abstract to attract the readers.

2.      Authors should add the skin sebum sampling method.

3.      Please mention the method of microphotography.

4.      Show how do you test the follicle and hair shaft state (fat and water presence).

5.      Determine the methodology used in measuring hair density and thickness clearly.

6.      How cotton soaked in diethyl ether absorbs hair sebum?

7.      How do you detect Scalp skin dryness?

8.      Where is the chemical analysis of these hair care products?

9.      Please support the manuscript with chemistry of hair care products.

10.  Authors should explain how oxidative stress causes alopecia.

11.  Authors could benefit from the following reference in the introduction: Chen, Y., Wan, X., Wu, D., Ouyang, Y., Gao, L., Chen, Z., El-Seedi, H.R., Wang, M.F., Chen, X. and Zhao, C., (2020). Characterization of the structure and analysis of the anti-oxidant effect of microalga Spirulina platensis polysaccharide on Caenorhabditis elegans mediated by modulating microRNAs and gut microbiota. International Journal of Biological Macromolecules, 163, 2295-2305.

Author Response

1.      Authors could add graphical abstract to attract the readers.

A1. We usually prepare Graphical Abstract to review papers, where mechanisms  of biomolecule effects are described. This is an original research paper, where the results are in focus and could attract a reader.

2.      Authors should add the skin sebum sampling method.

A2. The method is described in sub-section 2.5 of the Materials and Methods section

3.      Please mention the method of microphotography.

A3. The method of microphotography widely used in dermatology/trichology is described in sub-section 2.5 of the Materials and Methods section

4.      Show how do you test the follicle and hair shaft state (fat and water presence).

A4. The follicle state was determined by micro-trichological measurement of the follicle diameter: the bigger diameter the worse situation is due to the follicle swelling (Figure 3a). 

The hair shaft state was assessed by the visual analysis of microphotographs of hair shaft and skin taken under x200 magnification (Figure 3b). An expert in trichology/dermatology observed the presence of water and lipids on hair and scalp skin and quantified it in score.

5.      Determine the methodology used in measuring hair density and thickness clearly.

A5. The micro-trichology method is described in detail in sub-section 2.4 and illustrated by the Photographs (Figures  1 and 2 ). There are standard computer assisted calculations of either number of hairs per cm2 or single hair diameter in μm. “The instrumental method of the evaluation was a quantitative video-registration performed by an ARAMO SG videocamera connected to a computer equipped with the Trichoscience ver. 1.6 program” cyt. from the 2.4 sub-section.

6.      How cotton soaked in diethyl ether absorbs hair sebum?

A6. The cotton soaked in diethyl ether and rubbed against scalp skin is a standard method of sebum collection from any part of the skin described elsewhere, for example, our previous publication: Kostyuk V, Potapovich A, Stancato A, De Luca C, Lulli D, Pastore S, Korkina L. Photo-oxidation products of skin surface squalene mediate metabolic and inflammatory responses to solar UV in human keratinocytes. PLoS One. 2012;7(8):e44472. doi:10.1371/journal.pone.0044472. Epub 2012 Aug 30.

7.      How do you detect Scalp skin dryness?

A7. Hair dryness was determined by two ways: (a) the instrumental approach using tricho-microphotographs (Figure 3b) taken at high optical resolution x200, when hydration of the scalp skin could be seen and assessed by an expert trichologist and (b) by visual assessment of the dryness by a doctor-trichologist, who expressed the opinion in scores (Table 1).

8.      Where is the chemical analysis of these hair care products?

A8. We added the content of chemical excipients in lotions and shampoos to the sub-section  (evidenced by yellow colour)

9.      Please support the manuscript with chemistry of hair care products.

A9. Please, see A8

10.  Authors should explain how oxidative stress causes alopecia.

A10. It has been generally accepted that the molecular drivers of practically all cellular events underlying alopecia, such as altered hair cycle and inhibited keratinocyte proliferation, local chronic inflammation of different origin, apoptosis, angiogenesis,  and microcirculation are either free radicals or free radical-driven products of excessive oxidation (oxidative stress). All reviews and original research papers we referred to in the Introduction and Discussion sections of the manuscript mention this redox-dependent mechanism as a leading one for hair loss. We attempted to collect these views from the point of view of possible sources for free radical overproduction in various types of alopecia (Figure 11). Due to this common opinion, all natural remedies for hair loss prevention, hair growth promotion, and hair quality improvement contain plant/algae polyphenols and other substances with well established antioxidant properties. Moreover, the fermentation process leads to tremendous enhancement of antioxidant properties of plant parts since it biochemically modifies/liberates moieties with antioxidant potential. 

The discussion of the pathogenic role(s) of oxidative stress in different types of alopecia was beyond the scope of this original research. However, we are currently working on the review dedicated to the problem.

11.  Authors could benefit from the following reference in the introduction: Chen, Y., Wan, X., Wu, D., Ouyang, Y., Gao, L., Chen, Z., El-Seedi, H.R., Wang, M.F., Chen, X. and Zhao, C., (2020). Characterization of the structure and analysis of the anti-oxidant effect of microalga Spirulina platensis polysaccharide on Caenorhabditis elegans mediated by modulating microRNAs and gut microbiota. International Journal of Biological Macromolecules, 163, 2295-2305.

A11. We thank you very much for the reference of importance. We added it (numbered 64) and several other references (65-66) to the list of references (in yellow colour). We also added the last paragraph to the Discussion mentioning the great potential of marine plants-derived biomolecules as active cosmetic ingredients for hair care, hair loss preventing, and growth promoting.